# Eco-Innovation and Its Influence on Renewable Energy Demand: The Role of Environmental Law

**DOI:** 10.3390/ijerph20043194

**Published:** 2023-02-11

**Authors:** Muhammad Afaq Haider Jafri, Huizheng Liu

**Affiliations:** College of Economics and Management, Beijing University of Technology, Beijing 100124, China

**Keywords:** environmental law, innovation, education, renewable energy consumption

## Abstract

There is a consensus among the empirics regarding the positive role of renewable energy in mitigating the effects of climate change. Hence, it is vital to search for the factors that can promote renewable energy demand. As a result, this analysis investigates the impact of educational attainment, environmental law, and innovation on renewable energy consumption (REC) in China. From empirical estimates, we confer that the long-run estimates attached to the environment-related taxes and environmental policy stringency are positive and significant, implying that both these factors increase the REC in China in the long run. Similarly, the estimated coefficients of environment-related technologies and patent applications are significantly positive, confirming that environmental and other technologies give rise to REC in the long run. Likewise, the long-run estimates of education are significantly positive in both models, which confer that REC increases along with an increase in average years of schooling. Lastly, the estimates of CO_2_ emissions are significantly positive in the long run. These results imply that policymakers should invest in research and development activities that are crucial for promoting eco-innovation and renewable energy demand. In addition, strict environmental laws should be introduced to induce firms and businesses to invest in clean energy.

## 1. Introduction

Environmental pollution is one of the extensively-debated issues throughout the globe. Fast-growing economic development poses larger challenges for the environment. The large expansion of energy demand, rapid economic development, industrialization, population expansion, and increased utilization of natural resources caused disastrous challenges for environmental quality. Moreover, it is claimed that greenhouse gas emissions, specifically CO_2_ emission is a major contributing factor to environmental pollution. In order to address the negative externalities of environmental pollution, worldwide economies are formulating policies in the framework of the Paris Climate Agreement of 2015. To obtain the targets of the Paris Climate Agreement and to solve the issues of environmental degradation, economies are formulating several strategies to alleviate CO_2_ emissions [1]. It is believed that REC produces fewer CO_2_ emissions as compared to fossil fuel energy sources [2]. Additionally, REC significantly flourishes the economy [3]. There is quite extensive debate available in the literature regarding the significant positive contribution of REC in enhancing economic growth and alleviating CO_2_ emissions. Thus, it is important to investigate the dynamics factors of renewable energy demand. This research is moving in this direction to investigate the possible dynamics of renewable energy demand. Keeping in view the issues of climatic issues and global warming, economies throughout the world are adopting environmental-related technological innovations that are eco-friendly [4].

Environmental-related technological innovations stemming from a green investment in research and development help the economies to move towards sources of REC [5]. The findings of Ji et al. [6] for OECD nations on environmental innovation as a promoter of clean energy demand indicated that eco-innovation improves the share of renewables and decreases the utilization of carbon fuels. Ahmed et al. [7] also discovered that technical innovation helps polluted economies lower their carbon pollution. Additionally, Zhong et al. [8] concluded that China’s carbon footprint would decrease as a result of increased technological advancement. Khan et al. [9] discovered that ecological technologies are among the most significant elements accountable for increased REC in the OECD countries.

Moreover, it is argued that education sector development and green investment in R&D significantly encourage REC at the macro and micro levels. At the micro-level, it is expected that highly educated people are more responsive to environmental quality as they use energy-efficient goods, thus consuming little energy [10]. At the macro level, due to high educational attainment, economies can opt for contemporary technologies that transform the country toward highly sustainable energy sources [11]. Educational attainment affects the consumption of energy through technological progress and income channels [12]. Moreover, according to endogenous growth theory, educational attainment is positively connected with economic growth, which in turn increases REC. Furthermore, educational attainment also stimulates technological innovation that is highly linked with REC [13]. Educational attainment also results in promoting the consumption of clean and green energy, thus confirming the positive association between educational attainment and REC.

The literature also suggests that educational attainment complements research and development that improves the efficiency of production and mitigates dirty energy consumption [14]. It is claimed that technological innovation can increase output and improve efficiency that triggers input demand, such as natural resources and energy, that result in increased CO_2_ emissions [15]. Environmental green innovation contributes significantly to controlling CO_2_ emissions [11]. Studies also denote that technological innovation can replace capital and labor investment, thus stimulating the demand for REC [15]. Usman et al. [16] emphasized that sustainable economic growth cannot be achieved by eco-friendly innovation. Thus, in order to achieve a sustainable environment, eco-friendly green innovation is also required with the growth of research and development and REC [17]. But, very limited work is done in this direction to incorporate eco-friendly green innovation as a factor for REC.

The continuous increase in environmental degradation, carbon emissions, and the triggering issues related to energy security has highlighted the importance of renewable energy resources such as biomass, solar, geothermal, and wind [18,19]. In this regard, the use of REC can improve environmental sustainability. Thus, in literature, an increase in supply and demand for renewable energy is emphasized as a vital tool for the alleviation of CO_2_ emissions, achievement of sustainable development, energy security, and improvement in the quality of the environment [15]. In order to expand REC, economies are formulating environmental laws and regulations in the energy sector. For example, environmental laws aimed at increasing REC have become a significant element for energy and climate policies [20]. In the context of this discussion, it is realized that an increase in REC is a better strategy for improving environmental quality [16]. Governments and policymakers can formulate environmental laws that consider the demand for REC [21]. Studies denote that supervision, implementation, and formulation of environmental laws are important for the attainment of desirable impacts from REC. Johnstone et al.’s [22] study documented that environmental rules and regulations and energy prices contribute significantly to determining the demand and supply of renewable energy sources. To summarize the connection between environmental laws and the use of clean energy, the literature now in circulation may be classified into two main ideas. The first school of thought holds that environmental rules encourage renewable energy demand since they push companies and individuals into using renewable energy [23]. The second position is that restrictions on human activity in the name of environmental protection reduce the use of renewable energy [24]. In order to establish a mutually beneficial relationship between renewable energy and technological advancement, it may be necessary to use market-based mechanisms to promote environmental legislation. Since the exact impact of environmental laws on renewable energy consumption is unknown, we need more empirical evidence to have more clear picture regarding the impact of environmental laws on renewable energy consumption. Thus, this study is a move in this direction to determine the role of environmental laws on REC.

In the existing literature, the impact of environmental and macroeconomic variables such as energy prices, financial development, trade liberalization, CO_2_, energy intensity, and economic growth on REC has been explored quite extensively. However, connecting REC with education attainment, environmental law, and innovation creates important analytical and empirical deficiencies. In this context, a lack of evidence on education attainment, innovation, and environmental law creates an imperfect understanding of REC. In order to fill the gap in the existing literature, being a pioneer study, this research aims to explore the impact of educational attainment, environmental law, and innovation on REC in China. The main question addressed by the paper is how educational attainment, environmental law, and innovation affect REC in China.

The study makes the following contribution to the field of the study while keeping in mind the above-stated literature gap. Firstly, the study investigates the impact of educational attainment, environmental law, and technological innovation on renewable energy in China from 1990–2019. Secondly, this is the first-ever study incorporating educational attainment, environmental law, and technological innovation in any country’s renewable energy demand function. Thirdly, in addition to the long-run analysis, the study also focuses on the short-run estimates. Fourthly, the research methods used by the analysis, such as the ARDL model, are best suited when capturing the dynamic impact of independent variables on the dependent one. Fifthly, as opposed to panel data which suffers from aggregation bias, the study relies on the time series analysis, which is free from such types of issues [25]. Sixthly, the study will help in designing more appropriate policies for educational enhancement and research and development that help in transforming the energy sector towards the use of renewable energy. Lastly, as China is the largest emitter of CO_2_ emissions and the world’s largest economy in the world the implications of the study will not help reduce some environmental burdens in China but also globally due to the large contribution of China to the global pollution level.

The remainder of the article is structured as follows: The model, materials, and research techniques are covered in Section 2. The findings of our study are presented in Section 3. The conclusion of the study and policy implications for renewable energy development are presented in Section 4.

## 2. Model, Methods, Data

Energy innovation, particularly the development of cleaner energy—renewable energy—is viewed as one of the practical answers to environmental problems. Research and development (R&D) spending, particularly in the renewable energy industry, is crucial to promoting renewable energy and achieving sustainability while meeting a nation’s energy needs. Research and development in renewables support cleaner energy, expand employment possibilities, and support the growth of new businesses [26]. Since the relationship between R&D and technological innovations is positive, R&D is vital for promoting green technological development [27,28]. Boosting innovation by transitioning the industry away from current energy-intensive manufacturing methods and toward a less energy-consuming technology can aid in decreasing reliance on fossil fuels and increasing reliance on renewable energy sources. That is why eco-innovation is supposed to lessen the usage of fossil fuels and more heavily emphasize the adoption of renewable sources of energy.

The link between environmental laws and renewable energy consumption can be explained in light of the following arguments. The negative consequences of pollution may be reduced with the use of strict environmental laws [29], which encourage the creation of low-carbon technologies while prohibiting those of “dirty” ones. Strict environmental legislation may provide greener energy production, and use opportunities [30] can help to mitigate climate change. The Porter Hypothesis [31] suggests that well-crafted environmental policy may stimulate economic expansion and new forms of invention, such as renewable energy. Supporters of the Porter Hypothesis argue that stringent environmental laws may encourage nations to shift away from dirty to cleaner forms of energy.

In light of the aforementioned facts, we assume that the leading factors of the REC are environmental regulations, green innovation, education, and environmental pressures. The baseline regression model is as follows:(1)$${\mathrm{REC}}_{\;t}={\;\varphi }_{0}+{\;\varphi }_{1}{\mathrm{ER}}_{t}+\varphi_{2}{\mathrm{GI}}_{t}+\varphi_{3}{\mathrm{Edu}}_{t}+\varphi_{4}{\mathrm{CO}}_{2,t}+\varepsilon_{t}\;$$
where REC is renewable energy consumption, ER is environmental regulations, GI is green innovation, Edu is Education, and CO_2_ is CO_2_ emissions. Since an upsurge in environmental regulations is expected to upsurge the demand for renewable energy, thus we expect an estimate of $\varphi_{1}$ to be positive. Green innovation encourages the use of renewable energy by reducing traditional energy consumption and infers $\varphi_{2}$ will be positive. Also, education provides awareness of clean energy consumption, thus an estimate of $\varphi_{3}$ is expected to be positive. Carbon emissions are negatively influencing the consumption of renewable energy and are a sign of $\varphi_{4}$ will be negative. From Equation (1) estimation, we can obtain just long-run estimates. For the estimation of short-run effects of environmental regulations and green innovation on REC, Equation (1) represent error correction models as given below:(2)$${\Delta \mathrm{REC}}_{t}={\;\varphi }_{0}+\sum_{k=1}^{n}\beta_{1k}{\Delta \mathrm{REC}}_{\;t-k}+\sum_{k=0}^{n}\beta_{2k}{\Delta \mathrm{ER}}_{t-k}+\sum_{k=1}^{n}\beta_{3k}{\Delta \mathrm{GI}}_{\;t-k}+\sum_{k=0}^{n}\beta_{4k}{\Delta \mathrm{Edu}}_{t-k}+\sum_{k=1}^{n}\beta_{5k}{\Delta \mathrm{CO}}_{\;2,t-k}+{\;\varphi }_{1}{\mathrm{REC}}_{t-1}+{\;\varphi }_{2}{\mathrm{ER}}_{t-1}+\varphi_{3}{\mathrm{GI}}_{t-1}+{\;\varphi }_{4}{\mathrm{Edu}}_{t-1}+\varphi_{5}{\mathrm{CO}}_{2,t-1}+\lambda .{\;\mathrm{ECM}}_{t-1}+\varepsilon_{t}$$

Equation (2) represents the linear ARDL model form as recommended by Pesaran et al. [32] and gives short- and long-term results concurrently. The results of the short run are characterized through the coefficients connected to the “Δ” variables, while the long run results are presented by $\varphi_{2}$ to $\varphi_{5}$ normalized on $\varphi_{1}$. On the other hand, estimates of the long run are supposed to be authentic as their co-integration is determined through a t-test or an F-test. The critical values for both tests are taken for granted by Pesaran et al. [32].

To date, many time series estimation methods have been designed [33], which are significant in finding the co-integrating relationship between the variables. However, the ARDL model of Pesaan et al. [32] is considered superior to all other time series co-integration techniques due to the following benefits. Firstly, as stated above, this method is suitable for providing short- and long-term results by estimating a single Equation (2), while other time series only provide long run results. Secondly, the same integration order or I(1) is necessary for the variables to be co-integrated in other time series techniques. Nevertheless, the ARDL model is applicable in the case of varying orders of integration, i.e., I(0), I(1), or an amalgamation of these. However, the ARDL technique fails to deliver in the case of the I(2) variable due to the invalidity of F-statistics for such variables [34]. Thirdly, the ARDL method is efficient in the case of limited time series observations as opposed to other time series techniques, which do not yield systematic findings in case of a small sample size. Lastly, this method may also identify some feedback effects amongst the variables, thereby decreasing the endogeneity as well as multicollinearity risk [35].

The study aims to examine the effect of educational attainment, environmental law, and innovation on REC in China from 1990 to 2019. Table 1 provides detail regarding descriptive statistics and sources of variables. The dependent variable REC is measured as total energy consumption from renewable and others, and the data is acquired from EIA. The study used two proxies to measure environmental law, namely environmental regulations (environmental-related taxes as a percent of total tax revenues) and environmental policy stringency. The study also used different proxy measures of innovation; these are green innovation (environment-related technologies) and technology (patent applications, residents). Education is taken in terms of the average years of schooling. The study used carbon dioxide emissions as a control variable. Data for environmental policy stringency, environmental regulations, and green innovation is taken from OECD, while for the remaining variables, data is extracted from the World Bank.

## 3. Results and Discussion

Before performing empirical testing, it is mandatory to check the unit root properties of variables. For that purpose, the study embraced PP and DF-GLS approaches. In Table 2, all the variables are stationary at I(0) according to the PP unit root test. However, according to the DF-GLS approach, REC and green innovation are stationary at the level, and all other variables show non-stationarity at the level. Thus, they become I(1) stationary. On the basis of these findings, we decided to apply the ARDL approach to assessing empirical findings. Table 3 shows ARDL estimates. We have regressed two separate ARDL models because the study uses two proxy measures for environmental law and innovation. Model 1 reports the effect of ER, GI, and Edu on REC, while model 2 discloses the effect of EPS, Tech, and Edu on REC in China.

In the long run, the results of model 1 report that environmental regulations, green innovation, and educational attainment impact on REC are positive at 1%, 10%, and 5%, respectively. It reports that a 1% upsurge in ER, green innovation, and educational attainment tends to upsurge REC by 0.963%, 1.078%, and 2.627%, respectively. These findings demonstrate that the promotion of environmental regulations, green innovations, and educational attainment are significant policy measures for enhancing REC in China. These findings infer that environmental laws promote investment in renewable energy-related sources, which in turn enhances REC. However, renewable energy sources are attached with higher costs as compared to fossil fuel energy sources, while strong environmental regulations and environmental stringency policies enhanced REC in advanced economies. Environmental regulation has a positive influence on REC, and then China may have the chance to attain environmental gains. The relationship between environmental regulations and REC can be determined through non-monetary incentives. Environmental regulations can stop the loosening of environmental policies and strengthen lobbying efforts to promote REC. The positive link between environmental regulations and REC is also supported by [24,36], who observed that green technologies are promoted by environmental policies and consequently increase renewable energy generation and consumption. Moreover, the works of [23] also align with our findings and suggest the positive role of environmental regulations on REC.

It is found that the impact of GI on REC is also positive in the long run. This result is empirically supported by Zhang et al. [17], which state that green innovations lead to a significant reduction in CO_2_ emissions. Hence, most economies are adopting renewable energy sources and eco-friendly green innovations to alleviate carbon emissions. Additionally, green innovation results in increasing REC which also results in reducing CO_2_. Our findings support the results of Alvarez-Herranz et al. [37] and Khan et al. [38], which also demonstrate that green innovations encourage economies to converge toward REC sources. Green innovations also moderate the cost attached to REC sources making convergence from dirty energy sources to clean energy sources easy and convenient for economies [5]. Increased green innovation can also boost REC.

Findings show that education exerts a positive influence on REC in China. It might be due to the reason that educational attainment switches the economy towards more sustainable energy sources like REC, which also stimulate environmental innovations in the country. Moreover, educated labor is using sources of renewable energy. Our findings support the empirical outcomes of Yao et al. [39]. Broadstock et al. [40] establish that educated masses have more knowledge about the environment and thus choose efficient energy sources that consume less energy. However, Yao et al. [39] report that educational attainment results in rising clean and REC.

The relationship between CO_2_ and REC is significant and positive at the level of 5 percent in the long run. It states that a 1% expansion in CO_2_ emissions increases REC by 2.984% in the long run. Increasing environmental pressure in recent years has rapidly increased REC in China. This finding is backed by Uzar [41], who supposes that a dangerous level of CO_2_ emissions can force China to increase REC. Increased environmental regulations lead to an improvement in green economic activities by increasing REC. Thus results of our study show that environmental regulations, environmental innovation, education, and environmental pressure are vital in encouraging REC. In the short run, the findings of model 1 denote that environmental regulation, green innovation, and educational attainment increase REC at a 1% level of significance. CO_2_ is also stimulating REC in the short run.

The findings of model 2 demonstrate that EPS influence on REC is significant and positive at 5% in the long run. It states that a 1% increase in enforcement of EPS increases REC by 5.732% in the long run. Technological innovations have an insignificant impact on REC in the long run. However, the relationship between educational attainment and the REC is significant and positive at 5% in the long run. It discloses that a 1% increase in educational attainment enhances REC by 2.204% in the long run. It infers that educational attainment and enforcement of EPS are significant policy measures in China for the enhancement of REC. The study reports a positive link between CO_2_ emissions and REC in the long run. It reveals that a 1% rise in CO_2_ emissions tends to raise REC by 2.246% in the long run. The findings display that EPS and educational attainment are positively associated with REC at 10% and 5% levels in the short term, respectively. However, technological innovation again reports an insignificant impact on REC in the short run. The association between CO_2_ emissions and REC is significant and positive at the 5% level in the short run.

Findings of diagnostic tests are also reported. These tests are required for the validation of ARDL coefficient estimates. The F and ECM both tests approve the co-integration relationship among variables in both models. The absence of conventional time series problems is confirmed by the coefficient estimates of BP and LM tests in both models. The normality of error terms is confirmed by the RESET test in both models. The validity of estimates is confirmed by employing both CUSUM tests in both models. In Table 4, the bidirectional causal relationship exists between ER and REC. While unidirectional causality also exists from GI to REC and EDU to REC.

## 4. Conclusions

Over the past few decades, environmental degradation due to anthropogenic activities has become a major concern of international leaders because it has jeopardized the presence of humanity on earth. One of the most significant causes of environmental degradation and the resulting global warming is CO_2_ emissions because of heavy reliance on non-renewable energy resources. However, non-renewable energy sources have also contributed a lot in pacing the economic growth of the country. Therefore, policymakers and empirics are focusing on the carbon-free determinants of economic growth that can also preserve the environment. Literature on the factors of environmental quality suggests that renewable energy sources can protect the environment. However, very few studies in the existing literature have tried to analyze the elements that can affect REC. Therefore, we try to investigate the influence of educational attainment, environmental law, and innovation on REC in China by using the ARDL model.

From empirical estimates, we confer that the long-run estimates attached to the environment-related taxes and environmental policy stringency are significantly positive, implying that both these factors increase the REC in China in the long run. Similarly, the estimated coefficients of environment-related technologies and patent applications are significantly positive, confirming that environmental and other technologies give rise to REC in the long run. Likewise, the long run estimates of education are significantly positive in both models, which confer that REC increases along with an increase in education. Lastly, the estimates of CO_2_ emissions are significantly positive in the long run. In the short run, the results are mixed and inconclusive, to say the least.

Our results are important for policymakers because they can take the guidelines to make environment-related policies on the basis of these results. On one side, environment-related taxes and strictness in environmental policy may increase REC. On the other side, such stringency in environmental policies and a rise in ecological taxes may negatively impact the production of firms and businesses and increase their costs of production, which can affect the competitive position of the firms domestically and internationally. Therefore, policymakers should take much care while implementing such policies, and a balance should be maintained so that the nation’s total output is not affected. Similarly, policymakers should focus on green investment that promotes green technologies that will give rise to renewable energy sources. Further, increasing the literacy rate should also be part and parcel of any environmental policy because it can create awareness among the people about preserving the environment. As a result, people will start consuming more renewable energies.

Despite several significant contributions, this research has a few drawbacks that must be addressed in the future. For instance, since the research primarily considers China, its conclusions can only be generalized to emerging nations. Researchers should thus focus on estimating the relationship between environmental laws, green innovations, and REC in the context of developed and advanced economies in the coming times. Moreover, the current study uses the ARDL approach that can only capture the symmetric impact of environmental laws and green innovations on REC and overlook the asymmetric impact. Hence, future studies should apply the Non-linear ARDL and QARDL models that can also capture the asymmetric impacts.

## Figures and Tables

**Table 1 ijerph-20-03194-t001:** Data description.

Variables	Mean	Std. Dev.	Min	Max	Description	Sources
REC	6.326	5.218	1.483	17.63	Total energy consumption from renewables and other	EIA
ER	3.225	1.763	0.200	6.360	Environmentally related taxes, % total tax revenue	OECD
EPS	1.085	0.613	0.520	2.160	Environmental policy stringency	OECD
GI	9.795	1.508	7.387	11.99	Environment-related technologies	OECD
Tech	12.11	1.516	9.836	14.24	Patent applications, total	WDI
Edu	11.49	1.998	8.900	14.60	Average years of schooling	WDI
CO2	15.59	0.505	14.87	16.25	CO_2_ emissions	WDI

**Table 2 ijerph-20-03194-t002:** Unit root testing.

	PP			DF-GLS		
	I(0)	I(1)	Decision	I(0)	I(1)	Decision
REC	−0.325	−2.948 *	I(1)	−2.285		
ER	−1.715	−3.933 ***	I(1)	−1.022	−1.945 *	I(1)
EPS	−0.852	−4.235 ***	I(1)	−1.023	−4.235 ***	I(1)
GI	−0.201	−2.658 *	I(1)	−1.654 *		
Tech	−0.652	−4.302 ***	I(1)	−0.522	−3.889 ***	I(1)
Edu	0.321	−3.156 **	I(1)	−0.425	−2.856 ***	I(1)
CO2	−0.589	−3.125 **	I(1)	−0.278	−3.152 ***	I(1)

Note: *, ** and *** denote 10%, 5% and 1% level of significance, respectively.

**Table 3 ijerph-20-03194-t003:** Short and long run estimates of ARDL.

	Model (1)				Model (2)		
	Coefficient	S.E	t-Stat	Variable	Coefficient	S.E	t-Stat
**Short run**							
D(ER)	0.197 *	0.112	1.758	D(EPS)	0.905 *	0.521	1.736
D(ER(-1))	−0.057	0.175	0.326	D(EPS(-1))	0.607	0.460	1.320
D(ER(-2))	0.178	0.122	1.464	D(TECH)	0.339	0.398	0.850
D(GI)	1.832 *	1.102	1.662	D(EDU)	2.009 **	0.849	2.365
D(GI(-1))	−2.339 **	1.099	2.128	D(EDU(-1))	1.995 **	0.780	2.558
D(EDU)	1.263 *	0.719	1.756	D(CO2)	1.843 **	0.831	2.217
D(EDU(-1))	−3.236 ***	0.885	3.657	D(CO2(-1))	−5.879 **	2.289	2.568
D(CO2)	1.457 *	0.758	1.922	D(CO2(-2))	7.188 ***	2.307	3.116
D(CO2(-1))	−3.283 *	1.812	1.811				
D(CO2(-2))	7.117 ***	1.810	3.932				
**Long run**							
ER	0.963 ***	0.110	8.775	EPS	5.723 **	2.733	2.094
GI	1.078 *	0.626	1.723	TECH	2.266	2.469	0.918
EDU	2.627 **	1.083	2.543	EDU	2.204 **	0.981	2.244
CO2	2.984 **	1.314	2.270	CO2	2.246 *	1.201	1.870
C	5.627 ***	1.183	4.757	EDU	16.20 ***	5.581	2.904
**Diagnostics**							
F-test	11.18 ***				8.671 ***		
ECM(-1)	−0.673 ***	0.240	2.809		−0.450 **	0.076	5.921
LM	2.135				1.758		
BP	1.654				0.681		
RESET	1.542				0.123		
CUSUM	S				S		
CUSUM-sq	S				S		

Note: *** *p* < 0.01; ** *p* < 0.05; * *p* < 0.1.

**Table 4 ijerph-20-03194-t004:** Results of causality test.

Null Hypothesis	F-Stat	Prob.	Null Hypothesis	F-Stat	Prob.
ER → REC	10.60	0.001	EPS → REC	1.197	0.323
REC → ER	6.363	0.007	REC → EPS	1.132	0.342
GI → REC	8.201	0.003	TECH → REC	6.797	0.006
REC → GI	1.390	0.272	REC → TECH	0.964	0.398
EDU → REC	9.726	0.001	EDU → REC	9.726	0.001
REC → EDU	0.089	0.915	REC → EDU	0.089	0.915
CO_2_ → REC	10.69	0.001	CO_2_ → REC	10.69	0.001
REC → CO_2_	0.038	0.963	REC → CO_2_	0.038	0.963
GI → ER	1.794	0.192	TECH → EPS	2.183	0.139
ER → GI	1.298	0.295	EPS → TECH	0.796	0.465
EDU → ER	3.133	0.066	EDU → EPS	3.794	0.040
ER → EDU	0.851	0.442	EPS → EDU	0.512	0.607
CO_2_ → ER	2.418	0.115	CO_2_ → EPS	3.274	0.059
ER → CO_2_	1.369	0.277	EPS → CO_2_	0.515	0.605
EDU → GI	0.332	0.721	EDU → TECH	0.137	0.873
GI → EDU	3.955	0.036	TECH → EDU	2.669	0.094
CO_2_ → GI	0.324	0.727	CO_2_ → TECH	0.724	0.497
GI → CO_2_	2.883	0.079	TECH → CO_2_	1.949	0.169
CO_2_ → EDU	4.854	0.019	CO_2_ → EDU	4.854	0.019
EDU → CO_2_	2.318	0.124	EDU → CO_2_	2.318	0.124

## Data Availability

All data generated or analyzed during this study are included in this published article.

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
