# Peer review of "Eco-Innovation and Its Influence on Renewable Energy Demand: The Role of Environmental Law"

_ijerph, 2023, doi:10.3390/ijerph20043194_

Round 1
Reviewer 1 Report
It would be better if more discussions (maybe one or two paragraphs) on the theoretical framework for empirical models are included in Section 2. This could be done by using some of the literature review in Section 1.
1. What is the main question addressed by the research?
The main question addressed by the paper is how educational attainment, environmental law, and innovation affect renewable energy consumption (REC) in China. The paper empirically investigates the issue with data over the period 1990-2019.
2. Do you consider the topic original or relevant in the field? Does it address a specific gap in the field?
Yes. The topic is quite interesting and the paper uses the most recent data on the issue.
3. What does it add to the subject area compared with other published material?
The first contribution is using eco-friendly green innovation as an additional determinant of REC. The second one is the connection of REC with education attainment and environmental law. The third one is the data used that are most recent, and the measurements of the three indicators (educational attainment, environmental law, and innovation).
4. What specific improvements should the authors consider regarding the methodology? What further controls should be considered?
The theoretical framework for empirical models should be included.
5. Are the conclusions consistent with the evidence and arguments presented and do they address the main question posed?
Yes. The paper has done well in this aspect.
6. Are the references appropriate?
Yes. I don’t see any problems in References.
7. Please include any additional comments on the tables and figures.
No additional suggestions on the tables and figures.
Author Response
Manuscript ID: ijerph-2137615
Title: Eco-innovation and its influence on renewable energy demand: The role of environmental law
Dear Editor and Reviewers,
We would like to commence by thanking the editor and the reviewer for their valuable time and constructive comments. Their expert knowledge of the field has helped us to strengthen the manuscript significantly. According to the valuable suggestions provided by the reviewers, we have revised the manuscript. We endeavored to address all the comments and our reflections are now given below point by point. Changes to the manuscript are shown in red.
Sincerely
The Authors

Reviewer 2 Report
This study is “Eco-Innovation and its Influence on Renewable Energy 2 Demand: The Role of Environmental Law” is an interesting work however there are some issues need to be addressed before the paper accepted for publication in this journal.
1. I suggest to add a two lines background in the start of abstract in order to let the reader know what is the study going to investigate and what is the issue in this research.
Likewise, a two line recommendation at the end of abstract to be provided. I suggest to rewrite the abstract.
2. The introduction should add recent published papers from 2019 and onward. The innovation and contribution of the study should be further highlighted in the introduction. Likewise, the structure of the study should be provided at the end of introduction. There are some statements in the introduction that need the support of references.
3. To better show the research gap in the existing studies, I suggest the authors to add more recent work to literature review. Studies closely related to this work as well some other factors related to environment should be added to the literature review. I suggest few articles to be added to the literature as well the authors should download more recent published papers and organize the literature review.
4. Methodology need the support of references and its suitability should be explained for this study.
5. Results should be supported with references
6. The conclusion needs to be directly derived from the results. The limitation and future research direction should be highlighted further .
Author Response

(The authors gave the same response as above.)
